# Neural Networks in the Design of Molecules with Affinity to Selected Protein Domains

**DOI:** 10.3390/ijms24021762

**Published:** 2023-01-16

**Authors:** Damian Nowak, Rafał Adam Bachorz, Marcin Hoffmann

**Affiliations:** 1Quantum Chemistry Department, Faculty of Chemistry, Adam Mickiewicz University in Poznan, Uniwersytetu Poznanskiego 8, 61-614 Poznan, Poland; 2Institute of Medical Biology of Polish Academy of Sciences, Lodowa 106, 93-232 Lodz, Poland

**Keywords:** machine learning, neural networks, molecular docking, RORγ, drug design, SELFIES

## Abstract

Drug design with machine learning support can speed up new drug discoveries. While current databases of known compounds are smaller in magnitude (approximately $10^{8}$), the number of small drug-like molecules is estimated to be between $10^{23}$ and $10^{60}$. The use of molecular docking algorithms can help in new drug development by sieving out the worst drug-receptor complexes. New chemical spaces can be efficiently searched with the application of artificial intelligence. From that, new structures can be proposed. The research proposed aims to create new chemical structures supported by a deep neural network that will possess an affinity to the selected protein domains. Transferring chemical structures into SELFIES codes helped us pass chemical information to a neural network. On the basis of vectorized SELFIES, new chemical structures can be created. With the use of the created neural network, novel compounds that are chemically sensible can be generated. Newly created chemical structures are sieved by the quantitative estimation of the drug-likeness descriptor, Lipinski’s rule of 5, and the synthetic Bayesian accessibility classifier score. The affinity to selected protein domains was verified with the use of the AutoDock tool. As per the results, we obtained the structures that possess an affinity to the selected protein domains, namely PDB IDs 7NPC, 7NP5, and 7KXD.

## 1. Introduction

Designing a molecule that can effectively bind to a target protein domain is essential in the drug discovery process [1,2]. Computational methods can speed up the screening in a virtual manner, which helps to reduce excessive costs and the length of time necessary during the conduction of experimentally based techniques (so-called in-vitro or in-vivo studies) [3].

The use of a neural network may help speed up the acquisition of molecules that are similar to the known molecules of the desired properties. Artificial intelligence enables obtaining new biologically active compounds derived from known molecules via modifications that are necessary to better fit the pharmacological purposes based on molecular descriptors [3].

Crystallography and multidimensional nuclear magnetic resonance (NMR) [4] provide structural information deposited in a protein data bank (PDB) [5,6] that can be used during a search for interactions between a newly designed potential drug and selected macromolecules.

Currently, the methods employed by the most popular programs assume the flexibility of a ligand (a small molecule), and the rigidity of a receptor. Such an approach leads to a cost and time reduction. Programs, such as AutoDock [7], FleX [8], DOCK [9], GOLD [10], ICM [11], Glide [12], Ligand Fit [13], and others, bind small molecules to proteins [14]. Molecular dynamics simulations can be used to analyze the time-dependent evolution of ligand–receptor complexes and provide tools that help in macromolecule relaxation. This approach is more computationally demanding and requires more computational power [14].

In recent decades, a variety of docking programs have been developed for either academic or commercial use (vide infra). Different solutions and strategies are exploited in the context of ligand placement in the protein environment. In principle, they can be divided into four categories: stochastic Monte Carlo (Glide), fragment-based (Surflex, FleX), evolutionary-based (GOLD, AutoDock), and shape-complementary methods (LigandFit) [14,15]. A systematic search is not used due to the impossibility of the exploration of all degrees of freedom during molecular docking procedures. It is due to enormous computational costs. For example, if one is going to examine the cubic active site of $10^{3}$ Å${}^{3}$ with a simple ligand, and when energy evaluation is done every 10° (change of the angle between the small molecule and receptor), as well as a rigid movement every 0.5 Å [14] for a drug with four rotatable bonds only, there are 6 ×$10^{14}$ [6] conformations to be checked. If our computer is fast enough to compute 1000 conformations per second, the whole procedure would take 19,025 years to complete the systematic approach [14].

The studies presented here are aimed at proposing new molecules that may be considerable ligands against the selected protein domains (RORγ). This was accomplished by combining artificial intelligence that handles new potential drug generation with a chosen molecular docking program, i.e., AutoDock [16], which is in charge of determining the efficiency of the new ligand’s binding to the chosen receptor.

The protein domains researched in this study belong to RORγ proteins. They are referred to as orphan receptors since their natural ligands are undetermined [17]. RORγ proteins are associated with several processes in our bodies, including metabolic regulation, whole-body development, cell apoptosis, homeostasis maintenance, and circadian rhythm modulation [18,19]. The biological relevance is that they are associated with a variety of human diseases, such as atherosclerosis, osteoporosis, autoimmune disorders, obesity, asthma, and cancer [20,21,22]. RORs are thought to be the key regulators of Th17 differentiation [23,24]. They can be found in the heart, liver, testis, and muscles [18,25].

The main aim of the study was to check the possibility of new chemical structure generations with the application of artificial intelligence—in this case, neural network architecture. The machine learning model should be trained on how chemical structures are constructed. When the model has some chemical knowledge, it can generate chemically correct structures. The output is the SMILES code of a molecule [26], which is extremely useful because it can easily be used in other in silico tasks [1,2].

The second goal is to perform molecular docking automatically. This solution leads to a more effective energy-binding calculation when the whole process can be done with a dedicated script. Thus, many potential ligands can be checked inside specific macromolecules and their interactions can be compared with each other [27]. The use of IT tools enhances molecular screening due to the avoidance of the necessity of the manual preparation of a ligand and a given receptor. This approach makes it more efficient, and many systems can be studied [27].

Indeed, it helps us in obtaining the best structures, which have the most favorable binding energies (lowest values). They may be later synthesized, and an experimental verification can be carried out [28]. Within the procedure briefly sketched above, the search for new drugs can be boosted, and the selection of structures that will most likely exhibit desired properties can be achieved.

The model may predict new chemical structures based on the initial structures. The model can be used in drug or chemical designs in general, depending on the training data. This strategy can be used to solve a variety of structural design problems.

## 2. Results and Discussion

### 2.1. Model—The Neural Network

The prepared sequence-to-sequence model can construct semantically correct structures. The model was trained with the use of 121,000 structures. The applied loss function, i.e., the categorical cross-entropy [29] (see Appendix A) shows that training progresses well and the loss value converges to 0 (see Figure 1).

The given model shows the possibility of learning chemistry via neural networks. Application of SELFIES codes leads to no errors during the prediction step. In the case of the application of SMILES chemical information, there is a possibility of the formation of incorrect structures [2,30]. This fact is due to the lower robustness of the SMILES notation in comparison to SELFIES [2,30]. This model can handle molecules whose representation in the SELFIES form is shorter than 65 characters.

The other approaches to de novo drug design include conditional recurrent neural networks [1], which produce more targeted output, fingerprints of known molecules using sequence-to-sequence reconstruction [31], and multi-layered gated recurrent units (GRU), as well as other RNN architectures [1,2,30]. In these approaches, the steering of what the neural network produces is different.

The main limitation of the used model is the maximal length of molecular sequence that can be effectively encoded. The neural network employed can handle up to 65 SELFIES characters, although there is no restriction to the maximum SELFIES length in general. The neural network’s input layer definition determines it. This approach can handle SELFIES representations of molecules of any length, however, dealing with lengthy representations will be computationally expensive. It is important to remember that the SELFIES length is related to the length of the molecular sequence. This leads to a possible limitation of the initial structure’s molecular sequence length. Although the loss and validation loss values may be lower as the number of training epochs increases, this newly created one allows one to search for a close molecular space, which can also be promising for new drug discoveries.

The neural network proposed here is incapable of distinguishing whether a specific structure is active or not. This could be performed by employing subsequent classifiers and molecular docking based on the biological activity of specific receptors, such as nuclear receptors.

### 2.2. Prediction Initializers

By the choice, the 36 structures gathered by Y. Zhang, et al. [20] (see Appendix A) were selected. After the conversion, the stereochemistry of resulting SMILES codes was ignored. The distributions of the first and second selections are given below (see Appendix A).

The 21 structures passed through the QED selection and 9 passed through Lipinski’s rule of 5. The SYBA score was calculated for each of the previously selected structures (see Appendix A). The “My score” for structures that passed the SYBA threshold was calculated (see Appendix A).

### 2.3. Predictions and Results of Selection

With 0.1 tensor scaling, 55 distinct structures were generated. Only three of them were found in PubChem. Their CIDs were: 16445174, 18006105, and 129773833. The QED descriptor calculation results (see Appendix A), along with Lipinski’s rule of 5 fulfillment (see Appendix A), are shown below. Of these unique structures, 34 met the QED requirement, and 31 met the second discriminator. The number of structures that met both criteria is equal to 26, or 47.27% of the generated compounds.

Using the 0.2 tensor scaling, 78 distinct structures were generated. Only three of them were found in PubChem. Their CIDs were again: 16445174, 18006105, and 129773833. The QED descriptor calculation results (see Appendix A), along with Lipinski’s rule of 5 fulfillment (see Appendix A), are shown below. Of these unique structures, 39 met the QED requirement, and 30 met the second discriminator. The number of structures that met both criteria was equal to 25, or 32.05% of the generated compounds.

Then 26 species from the first prediction and 25 from the second were combined, and repetitions were removed. It resulted in 42 unique structures. The third discriminator was applied—the SYBA classifier (see Appendix A). The results of its application resulted in 20 distinct structures (see Table 1).

As many as 11 out of 20 structures come from the first structure’s tensor scaling (see Table 1). This fact shows that the prediction given by this model is connected to the type of initial structure. After the removal of structures that cause some problems during the molecular docking procedure, the number of molecules reduces to 16 (see Table 2).

Compounds that “survived” the selection possess a generally high value in “My score” (see Appendix A), indicating that both the descriptor QED and SYBA scores are greater than zero. However, in two cases (molecule numbers ten and eleven), the lowest QED descriptor is maintained. In comparison with all the data after the first two selections, these structures go further due to a high SYBA score. Some structures possess the highest QED (molecule number fourteen) and SYBA score (molecule number eight) in comparison to all other structures generated after QED and SYBA selections.

All structures generated using the given method for potentially discovering new drugs were semantically valid. However, a new problem was discovered. a problem involving the possibility of exotic structure formation as well as some difficulties encountered while creating 3D structures. These difficult structures can be removed.

Most of the structures that are generated from the one that initially has high values of QED and SYBA score are above the threshold. This may lead to the conclusion that the initial structure has a significant role in the outcomes of the model.

### 2.4. Similarity to Initial Structures and Training Data

Structures from tensor scaling are not remarkably similar to initial structures (see Appendix A). One structure was replicated in the same shape. The lowest value of similarity was 0.17. Indicating that the AI-generated structure differs significantly from the initial structure.

Structures from tensor scaling are not highly similar to training molecules (see Appendix A). The highest similarity was found at 0.64, while the lowest similarity was 0.02; thus, the structures that give the results are different. The mean similarity is 0.26.

The comparability of structures that are selected for molecular docking gives information that these structures are also not identical (see Figure 2), and the thirteenth is the outliner as it has the lowest Tanimoto similarity output in each comparison.

When using the PubChemPy [32] library to see if the initial SMILES are present in PubChem, the following PubChem CIDs are returned: 807146, 16445174, 71470549, 71811962, and 135337558. These structures are related to the initial structures. Only one generated structure by the neural network was found in PubChem. All the above steps are performed in Appendix A. Questions can be raised about the similarity between structures after the first selection (42 objects) and the initial RORγ active set of compounds containing five structures (see Appendix A). The highest hit is 1.00, so the one recreated structure. The lowest one is 0.14. Appendix A shows the Tanimoto similarity between 42 structures and the training data (121,000 structures). As a result of Appendix A, neither the RORγ active dataset nor any generated structure appears in the training dataset.

Further preprocessing and selection of training data could be applied. More similar structures may be generated if training structures are built with a charset that is more closely related to the charset of target structures.

### 2.5. Molecular Docking of Selected Structures

The results of molecular docking are collected in Table 3. The lowest average binding energy was found for molecule number 12, which equals −9.80 kcal/mol. The second was molecule number 20, with −9.70 kcal/mol. The third was the 19th, with −9.60 kcal/mol. The minimal binding energy for the 7NPC—macromolecule was −10.0 kcal/mol, and the structure related to this result is molecule number 10. In the case of the 7NP5 −10.0 kcal/mol binding energy, it is found to be the lowest value, and the corresponding structures are the 7th, 19th, and 20th. The 7KXD complex, with a minimum energy of −10.0 kcal/mol, has the lowest energy, and molecules 11th, 12th, and 19th are the causes of it. The most favorable average binding energy is for molecule number 5, at −6.7 kcal/mol. It is the worst candidate for a new drug based on binding energy. For each of the chosen macromolecules, the less favorable binding energies are −6.5 kcal/mol (7NPC), −6.9 kcal/mol (7NP5), and −6.7 kcal/mol (7KXD), and molecules are related to those results in the order of 5 (7NPC), 5 (7NP5), and 14 (7NP5), 5 (7KXD), respectively (see Table 3).

In Table 4, the docking results are obtained for the re-docking procedure as we know the exact place of binding. A recreation of the initial pose was achieved in Appendix A with the use of AutoDockTools 1.5.6. [16] (Software creator: Morris, G.M., et al., source of the program: ccsb.scripps.edu, Country of origin: La Jolla, United States of America). In the case of molecular docking performed with help of the Python tool, the results are significantly different. They are as follows: −8.0 kcal/mol for the 7NPC ligand, −9.2 kcal/mol for the 7NP5 ligand, and −7.7 kcal/mol for the 7KXD ligand (see Appendix A).

The differences in the results are caused by the different sizes of the search space; they demonstrate how significant the search space size is. The time during which the genetic algorithm is allowed to search for the best poses is crucial, as it may lead to different results.

Karaś et al. demonstrated that molecular docking can be a powerful tool for identifying active compounds against macromolecular targets [35]. It was discovered that AZ 5104 is active against RORγ receptors [35].

Based on the study of D. Plewczyski, et al. [14], the tool that was used during the actual investigation failed in nearly 90 cases, while the total number of ligand–receptor pairs was 1300. AutoDock has successfully docked in approximately 93% of initial pairs. Computed ligand poses in the case of AutoDock led to poor results when compared with the initial pose of the molecule. The threshold, given in Å, was set at 2, and everything below it was marked as successfully docked. In this case, the tool utilized during this study was in last place with FleX [14]. According to D. Plewczyski, et al., AutoDock performs well in the case of small and hydrophilic molecules with either strong or weak binding energies (about 50% accuracy in top-scoring conformation-based analysis and about 76% accuracy in best-posing conformation-based analysis) [14]. Overall, the Pearson correlation for AutoDock, the top-scoring poses equal 25% and 19% in the best pose comparison). This could be due to the relative simplicity of the scoring functions and other assumptions that are made during virtual molecular docking. That is why the type of software provided is only a supporting tool in the drug design process [14].

Differences in energies after molecular docking procedures can be observed. It can be so due to the different requirements of each search. First, the so-called screening approach can be seen, in which many ligands are evaluated with many macromolecular systems. The second was to create visualizations in a bigger search space. This was done to check if the new structure would be attached to the same active site of the protein. It should be remembered that proteins possess different active sites. Some are for agonists, while others are for antagonists, so acting against these RORγ domains is not out of the question.

Molecular dynamics simulations (MDS) have a significant impact on molecular biology and the discovery of novel drugs. The study did not include this. It is a location for undeniable improvements to the method presented.

## 3. Materials and Methods

The three protein domains used as receptor targets (examples of the RORγ family of receptors) in this paper are 7NPC, 7NP5, and 7KXD, according to the PDB database infrastructure [5]. The choice was made according to the following requirements: they all belong to the RORγ family, they have similar and good data resolution, and each structure has one main chain; these domain data were collected using X-ray crystallography, and the publication year was 2021 [5,6,33,34].

All mathematical background equations, methods, and files, along with the additional figures, are presented in the supplementary data (see Appendix A) along with the additional figures. For the convenience of the reader, the whole project procedure is visualized in Figure 3. It reflects the proposed workflow involving a number of steps; in the following sections, we refer to the particular steps presented in the chart. The study starts with setting up the environment (see Stage 1, Figure 3).

### 3.1. Training Data

The compounds were extracted as SMILES code representations from the ZINC20 database [36]. The Appendix A contains the downloaded ZINC20 database (see Stage 2a, Figure 3).

This step consists of about 885,780,663 substances [37] within ca. 1800 tranches [37]. Then, in Appendix A, a selection of data that will be used to create a model can be found. This file contains information about the further tranches used. When the 935,475 unique isomeric SMILES are collected, they are translated into SELFIES [38], and the length of each SELFIES code is determined. It is assumed that only compounds with the SELFIES code length in the range of 30 to 50 are selected for further processing; this results in 569,205 structures. The length of SELFIES is proportional to the number of atoms in a specific structure.

The next step is to take advantage of the generic form of SMILES, which has a smaller charset in comparison to isomeric SMILES. As isomeric SMILES contain information about stereo centers and double-bond geometry, the charset is larger than in the case of generic SMILES, where the information is neglected.

As the SELFIES molecule representation is used in the neural network, it is necessary to convert SMILES into SELFIES. The charsets are different for the SMILES and respective SELFIES. The generic SMILES are translated into SELFIES codes and the lengths of the SELFIES are calculated.

Subsequently, the compounds containing rare heteroatoms (like Sn, Se, B, and P) were removed from the dataset (see Appendix A). The last effort is focused on the preparation of 121,000 structures for a model. This is achieved within Appendix A. The final structures that are our training data are collected in Appendix A. The data were then divided into training and validation sets using the sklearn library’s “train test split” functionality. There was no preparation of specific data for differentiation of the training and validation sets. Moreover, 10% of the acquired data set was used as the validation data.

The resulting distribution is presented in Figure 4.

### 3.2. Active Compounds against RORγ

Based on the publication by Y. Zhang et al. [20], RORγ active compounds were gathered (see Appendix A). Five structures were selected (see Table 5) to predict new structures. These structures were selected after the following steps (see Stage 2b, Figure 3).

As the training data are constructed based on generic SMILES codes (which are translated into SELFIES later), the collected data should also be converted into that form. The first step of selection (see Appendix A) means that the QED descriptor [39] (see SM Equation (S1)) is larger than 0.5. The second selection is done with the use of Lipinski’s rule of 5 [40] (see SM Equation (S2)), and only structures that fulfill the conditions are selected. Then the results are saved in Appendix A, and the third classifier, the SYBA algorithm [41,42], is used (see Appendix A). That one’s threshold is set to above 0. It decides if the structure is easier (higher value) or harder (lower value) to synthesize (see Appendix A). After that, the normalization of the QED descriptor and SYBA score are calculated.

The last calculated parameter, called “My score”, is an arithmetic mean of the normalized QED descriptor and SYBA score (see SM Equation (S3)). In the next step, the results are saved in Appendix A. The results are sorted in descending order of “My score”, and five structures are selected—two from the top, two from the middle, and the last one; these were selected for the predictions. They were saved in Appendix A.

### 3.3. SELFIES Coder

The neural network requires a mathematical representation of the data, in this case, molecules, to be fed into it. In order to vectorize SELFIES code, the SELFIES coder tool was developed. A vectorization procedure converts a text representation of a species into a one-hot encoded form that is readable by a computer. This procedure generates the set of molecules in a form suitable for machine learning training [43]. The functionality SELFIES coder (see Appendix A) has been prepared in order to convert the SELFIES codes [38] into molecular sequences and further into numerical vectors (vectorization).

Since the SELFIES [38] encoding of molecules is more robust compared to SMILES [26] (invalid SMILES can be formed); this approach assures that all structures are semantically correct, which is not the case with the SMILES encoding [1,2,30]. Each SELFIES unit is turned into a mono-character according to certain rules, which ultimately leads to a molecular sequence (composed of characters from Appendix A). This molecular sequence is later transformed into a one-hot encoded form within a vectorization procedure.

The SMILES format is very useful for Python molecular docking approaches, and it is effectively supported by popular libraries, such as RDKit or PubChemPy. The SELFIES format was investigated for neural network applications and to determine whether semantically incorrect SELFIES creation is possible.

### 3.4. Model—The Neural Network

The neural network is created (see Stage 2a and Stage 3a, Figure 3) in Appendix A. The basic idea was to prepare a model based on recurrent neural units that are capable of generating molecules from those of known activity. The idea of the model’s construction was taken from [44]. The recurrent neural network (RNN) applied here (see Appendix A) takes advantage of long short-term memory (LSTM) cells [45], which help in learning dependencies of sequential data (see Appendix A).

The data collected in the previous section (see Appendix A) are used to create the model. The 108,900 structures are used as training data, and 12,100 structures are used as validation data. The SMILES codes are translated into SELFIES, and from the SELFIES codes, the charsets are created (see Appendix A). Then the molecules are in the form of a so-called molecular sequence, and two additional charsets are created, one containing translations from arbitral mono-signs (single characters) to numbers (see Appendix A) and the other containing translations from numbers to arbitral mono-signs (see Appendix A).

Now the molecular sequences can be coded into latent space (see Appendix A). Vectorization is the process of converting molecular sequences into one-hot encoded arrays. Additionally, the characters reflecting the beginning and the end of the molecule were introduced namely: “!” as the starting character and “E” as the ending character. Moreover, the latter is used as a padding character ensuring the same length of the molecules. The maximum length of the molecular sequence is called the embed value. It determines how long molecular sequences can be used during predictions.

The other functionalities are saved in Appendix A. The first decodes states from latent spaces (see Appendix A) and the second is responsible for making (character by character) predictions (see Appendix A).

The data transformation scheme for the neural network is presented in Appendix A. The maximum epoch number was set to 200.

### 3.5. Predictions

The structure generation was carried out by tensor scaling and further converting molecular sequences back into SMILES (see Figure 5, see Stage 3b, Figure 3). It was initialized by previously selected structures—these are gathered in Appendix A (see Table 5). The saved models are used, as well as the charsets. The initial structures are then translated into molecular sequences and vectorized so that the model can make predictions. The function latent_to_mol_seq (see Appendix A) simply converts a latent space into the molecular sequence. As per the scheme, the states are taken from predictions made on newly obtained latent spaces and the states of the sample model are reset with new ones. Then a “for” loop makes character-by-character predictions until the “E” character is encountered, then it stops creating a molecular sequence.

Predictions are made around the initial latent vector of each initial molecule by adding random noises of varying amplitudes (0.1 and 0.2). For each initial molecule, 20 samples were taken, yielding 100 structures. Potential duplicates were then removed. As a result, molecular sequences were generated, which were then decoded into SELFIES and translated into SMILES. SMILES codes are converted to canonical form and searched in the PubChem database. These steps are shown in Appendix A, with 0.1 and 0.2 tensor scaling, respectively. The results are saved in Appendix A for 0.1 and 0.2 tensor scaling, respectively.

Based on Appendix A, the selection is made in Appendix A for each tensor scaling, respectively. If the QED descriptor [39] value is lower than 0.5, it acts as a discriminant (see SM Equation (S1), see Stage 4, Figure 3). Lipinski’s rule of 5 [40] is applied as the second discriminant (see SM Equation (S2)). The outcomes of both options were recorded in Appendix A. The outcomes of the merger of Appendix A are then saved in Appendix A. Each structure is assigned a prediction mode (see Appendix A), with the results saved in Appendix A.

The third selection step was applied (see Stage 5, Figure 3). It was done with the use of the SYBA classifier [41,42] (see Appendix A), resulting in Appendix A (see Appendix A). At an arbitrary value of 0, all species with an SYBA score less than this value were rejected from further consideration.

The structures that were selected can be viewed in Appendix A (see Stage 6, Figure 3). Immediately after this, molecular docking of selected structures could be applied (Appendix A). If some structures prove problematic during the docking procedure, they can be eliminated by rerunning the code present in Appendix A. Then molecular docking can be run one more time, but without problematic structures.

To analyze the potential drug-likeness of proposed structures, the QED and Lipinski’s rule of 5 classifiers were utilized. The third classifier checks the structure’s accessibility to synthesis. They are all employed before the molecular docking technique, assisting in the selection of the “best” structures for further testing.

The biological activity of the derived structures can be assumed based on the molecular docking technique and a partial resemblance to existing drugs with confirmed biological action. The technique suggested here could be thought of as a random search in the region of a specific species, such as one whose biological activity was experimentally determined. The changes incorporated into the species preserve the pharmacophoric properties of the original molecule to a considerable extent. Based on this assumption, it is reasonable to expect that the derived species will have similar characteristics. Nonetheless, experimental validation is the best technique to obtain a conclusion.

### 3.6. The Molecular Similarity

A molecular fingerprint is an abstract representation of some features of a given structure. It is a compact representation of chemical structure expressed as a series of binary digits. The molecular fingerprint used here is the “RDKit” fingerprint which is yet another implementation of a daylight-like fingerprint [46]. In the molecular fingerprint representation, the similarity of two species can be easily calculated [47,48]. In particular, the Tanimoto similarity coefficient involves two fingerprints and reflects the similarity of the molecules [49] (see Appendix A, Stage 7b, Figure 3). The value 1 represents identical molecules, while the value 0 indicates that no common components exist in the respective bit representations.

Firstly, a comparison between AI-predicted structures and the selected initial structures was made (see Appendix A). Then the Tanimoto similarity is calculated among the structures that are going to be docked. This means structures that meet the selection rules from the previous paragraph (see Appendix A). The third step is to compare training data with newly obtained compounds. The frequency axis has a much larger scale due to the number of compounds multiplied by the number of training structures (121,000). The results are displayed after calculations are performed for each docked structure and each training structure combination (see Appendix A).

After that, there is a simple check to see if the AI-generated structures can be found in PubChem (see Appendix A). The same procedure as described above is applied to all structures that meet QED and Lipinski’s rule of five thresholds (see Appendix A). Inside Appendix A (see Stage 8b, Figure 3), a check is performed to see if any of the RORγ active compounds are present in the database, as well as if any of the reconstructed entries are present.

### 3.7. Molecular Docking

The search for possible ligand–macromolecule interactions was investigated via a molecular docking procedure and with the utilization of the Pyscreener tool [27] (see Stage 7a, Figure 3). The more negative the binding energy, the better the affinity of the ligand to the receptor.

The procedure was conducted in Appendix A—it is an automated way of carrying out molecular docking within a Python script. As a result, a file is created (see Appendix A). Binding energies, structures, and coordinates are stored inside Appendix A, see Appendix A along with Appendix A. The results are extracted via the code present in Appendix A. We have used here the AutoDock Vina [50,51] program with the Lamarckian genetic algorithm [52].

The average binding energies are calculated based on molecular docking experiments with three different macromolecules (see Appendix A). Based on molecular docking experiments with three different macromolecules, the average binding energies are calculated (see Appendix A). The code in Appendix A then converts SMILES into a 3D structure that can be used for later manual molecular modeling. It is carried out by general chemistry rules concerning angles, hybridizations, and distances [53] (see Appendix A, (see Stage 8a, Figure 3)).

## 4. Conclusions

The model we proposed can create molecules that, in a limited manner, are similar to the training data and initial data. This can be due to the output formation–tensor scaling. This method takes advantage of more than one structure generation per single initial molecule as the input. The filters used (QED, Lipinski’s rule of 5, and SYBA score) enable us to obtain structures with good synthetic possibilities as well as low binding energies. There is still a necessity for confirmation by synthesis and experimental measurements if these structures have actual affinities for selected protein domains. Our solution can be applied to other macromolecules of interest as well, where at least one active compound is known. The possibility of a new chemical structure generation with the application of artificial intelligence was shown. The machine learning model has indeed gathered some chemical knowledge. The use of the Python code (see Appendix A) resulted in the automatized molecular docking procedure. The study shows the possibility of new molecule formation via a neural network that will exhibit mathematical affinity to the selected protein domains. Any known protein domain and any known chemical with activity against it can be used to replicate the process.

## Figures and Tables

**Figure 1 ijms-24-01762-f001:**
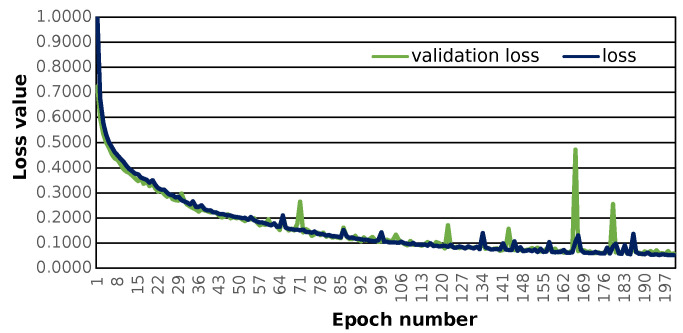
The training loss and validation loss during seq_to_seq model training. It follows the general rules of model training during time evaluation; both values are decreasing, indicating that the model is learning how to reconstruct the molecular sequence of the training molecules. It can be noticed that the loss value for the validation data also decreases, which means that the model learns to generalize.

**Figure 2 ijms-24-01762-f002:**
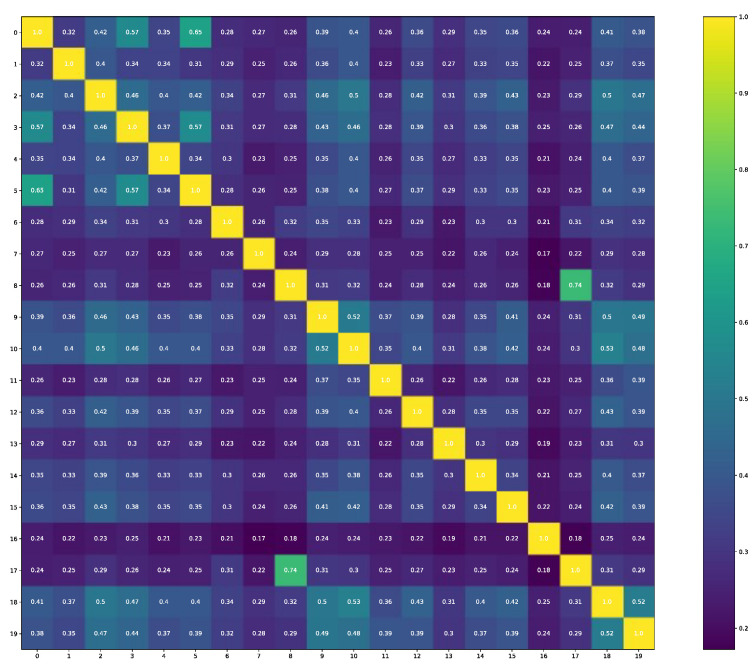
The Tanimoto similarity distribution along with the “to be docked” molecules.

**Figure 3 ijms-24-01762-f003:**
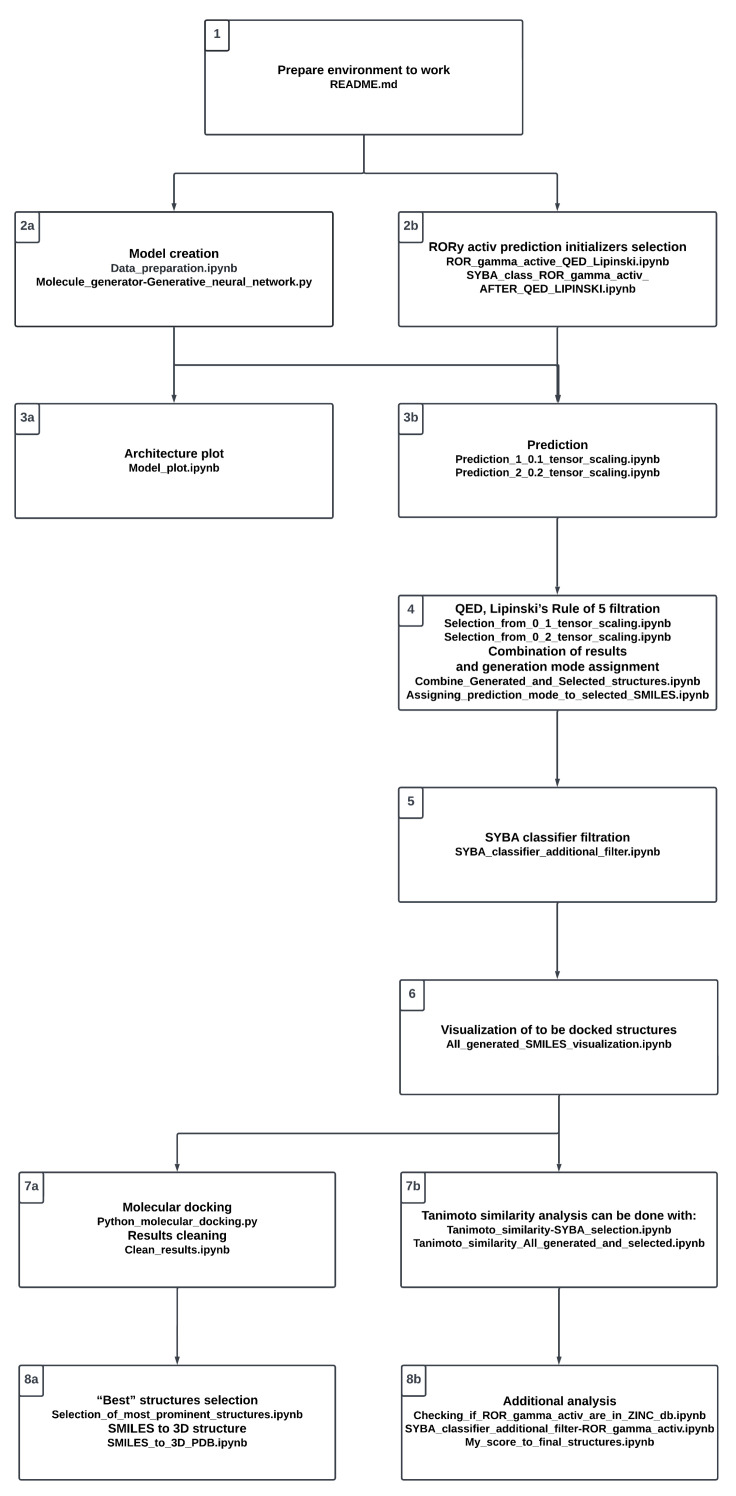
The flowchart of the overall workflow.

**Figure 4 ijms-24-01762-f004:**
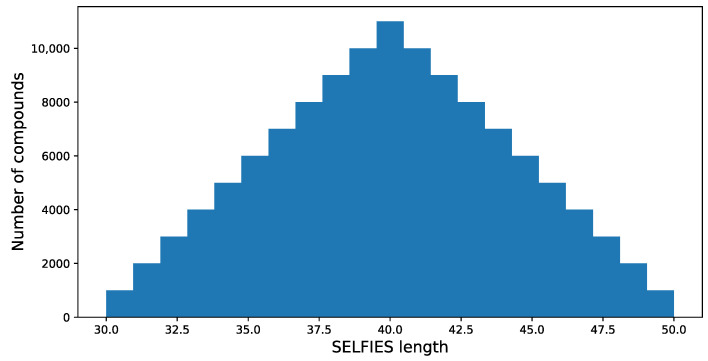
The training data SELFIES length distribution. As the training set, 121,000 structures are used.

**Figure 5 ijms-24-01762-f005:**
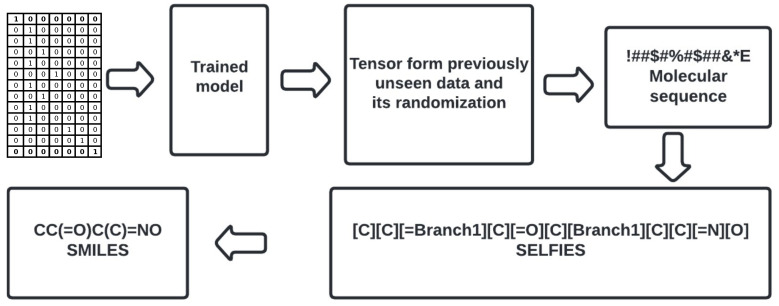
Workflow for data prediction.

**Table 1 ijms-24-01762-t001:** The structures of selected molecules. The table contains structure, structure number, structure of origin number, tensor scaling mode.

Newly Generated Structures’ Images			
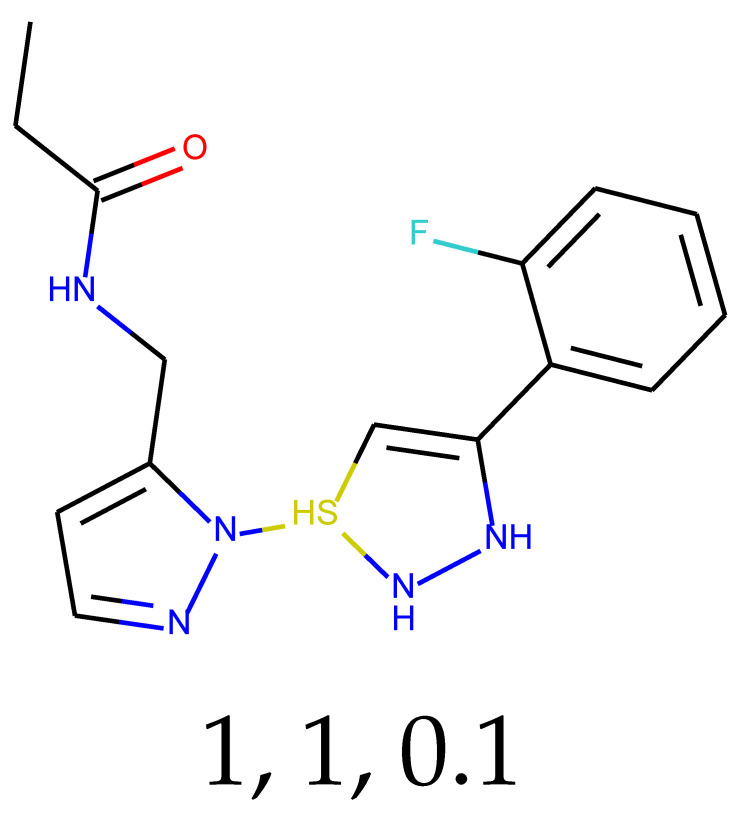	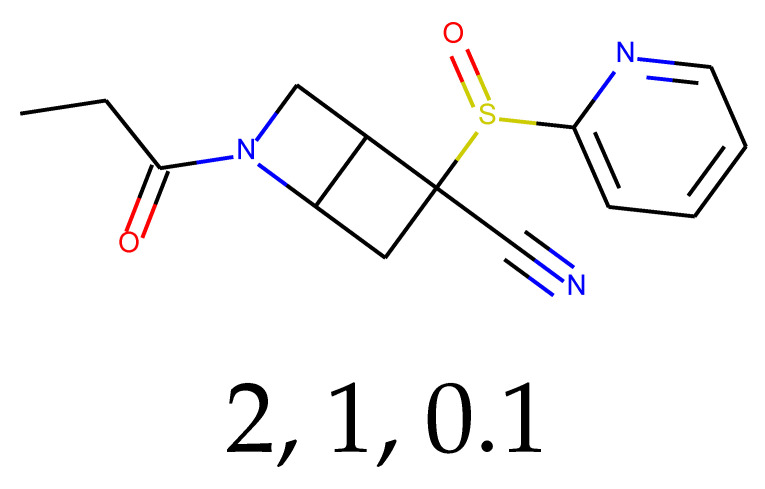	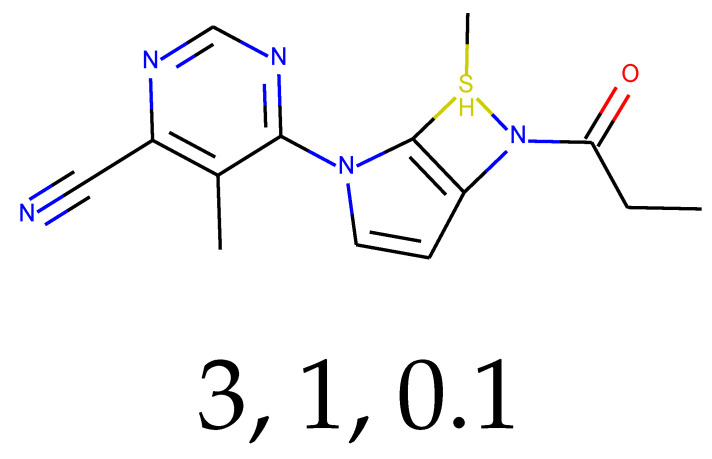	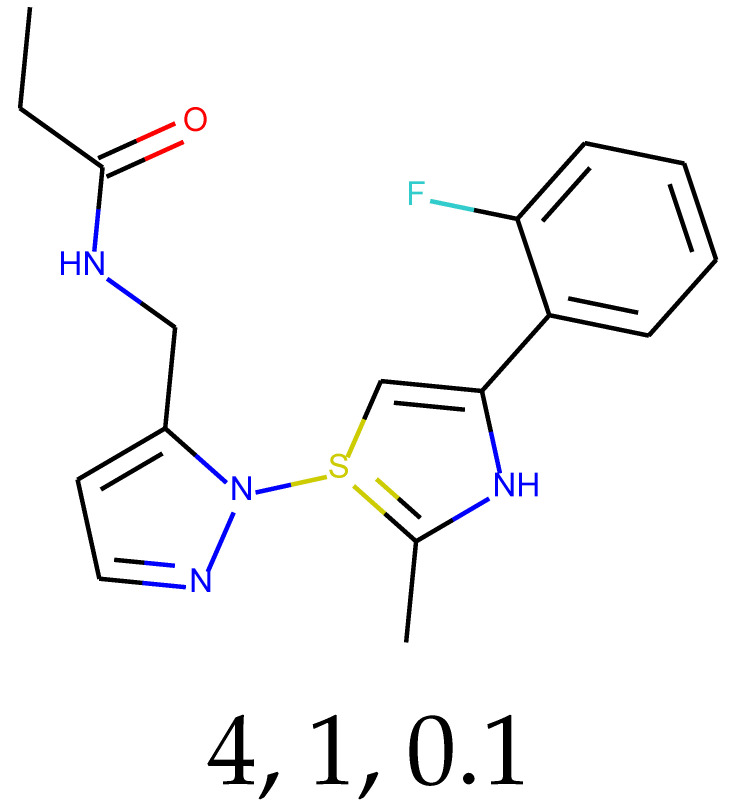
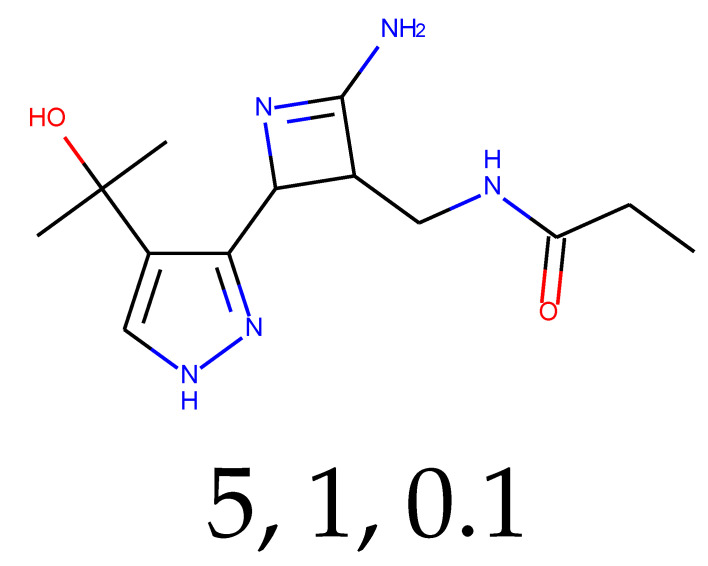	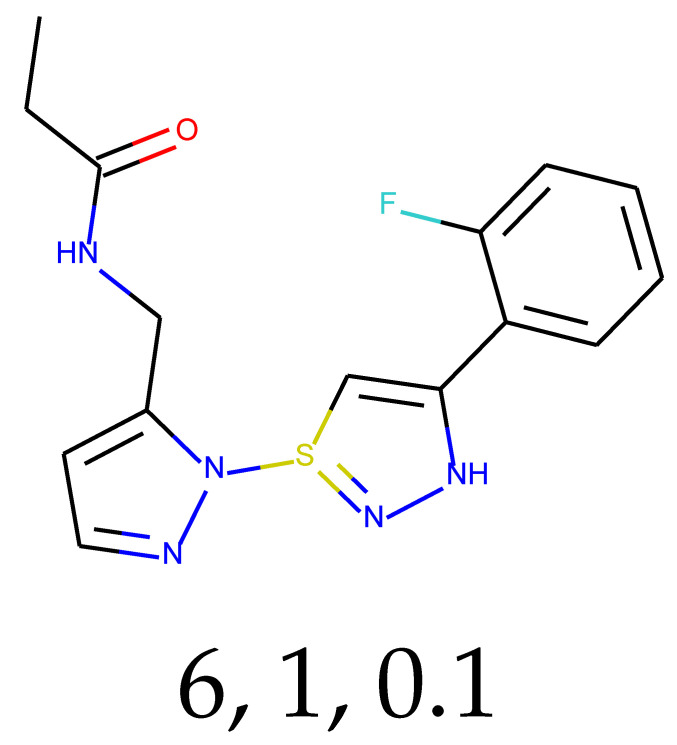	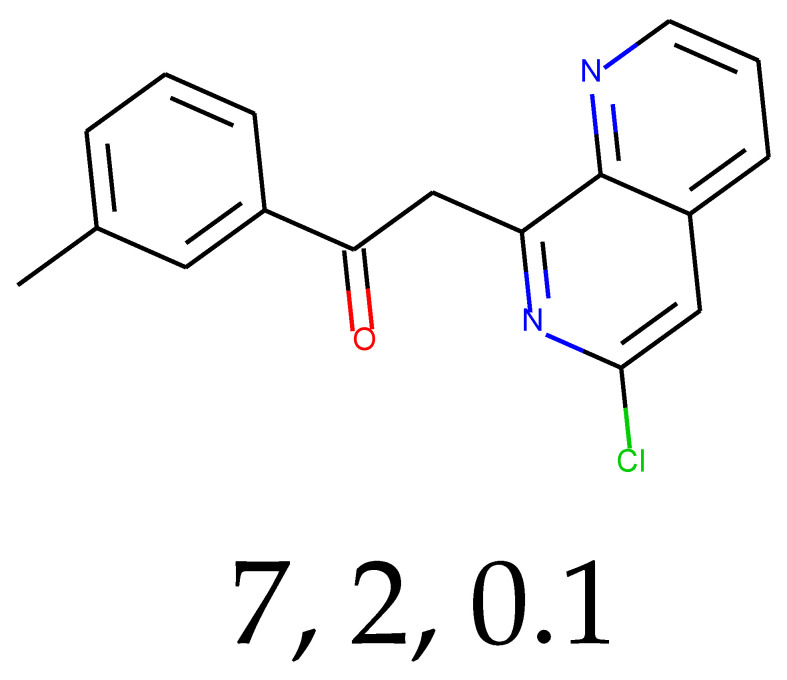	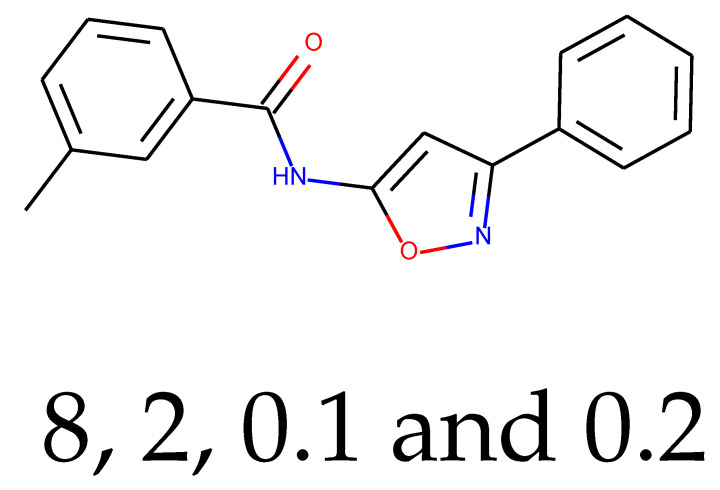
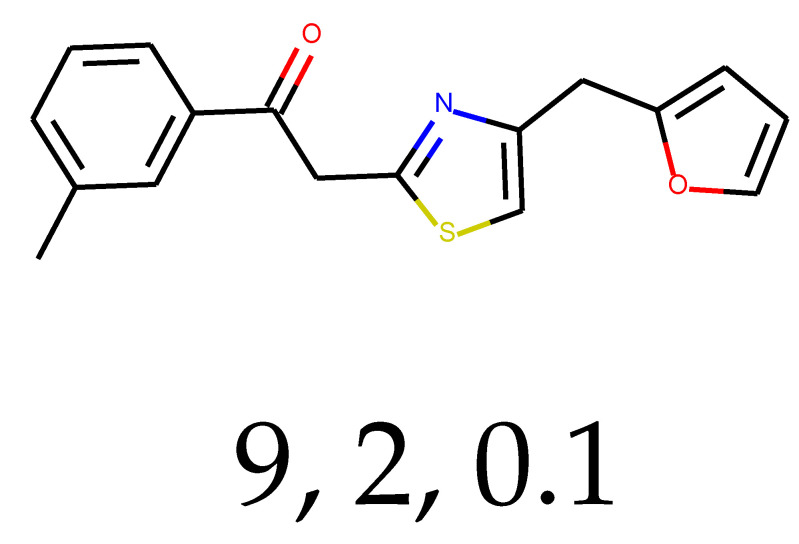	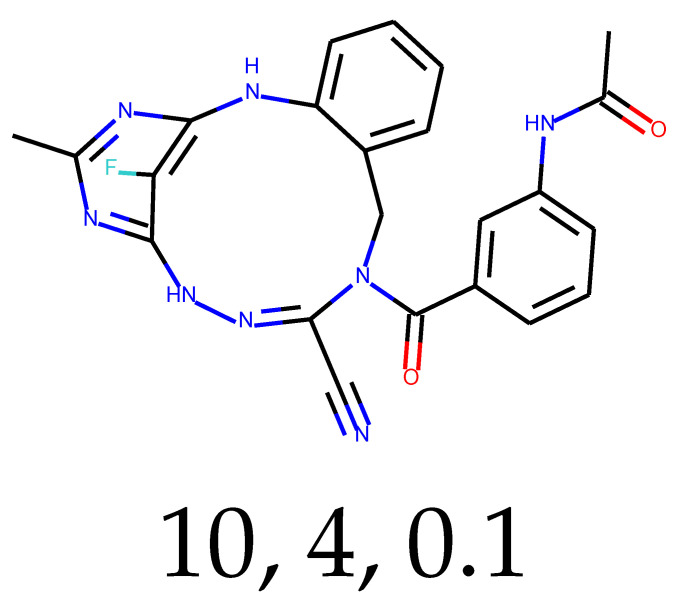	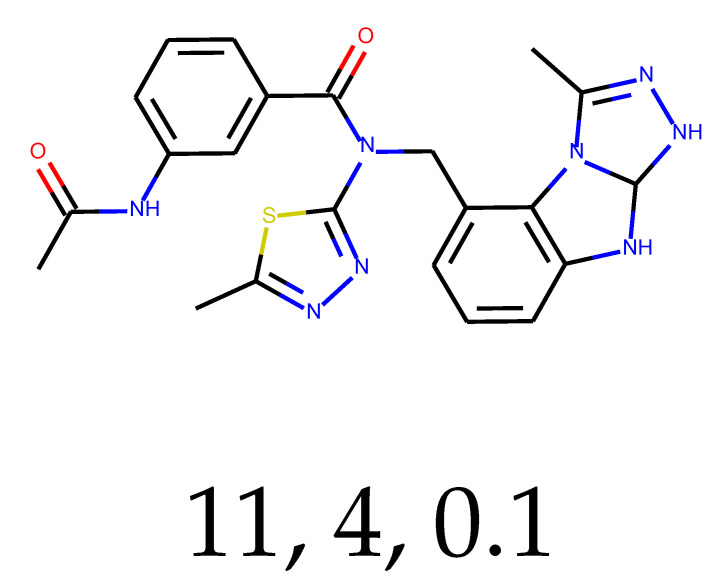	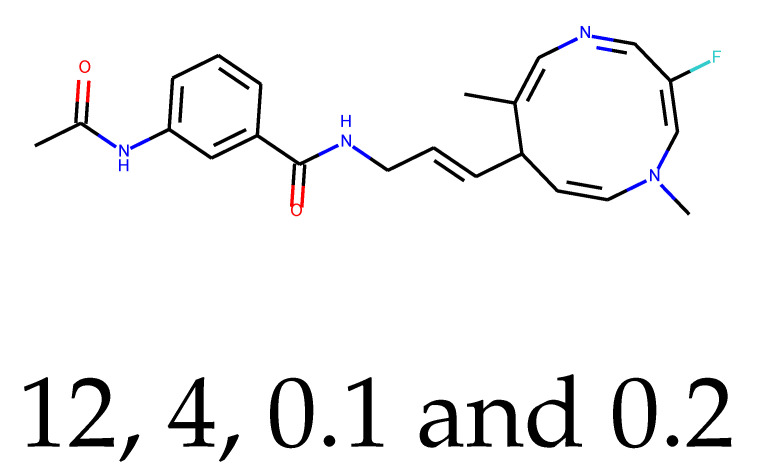
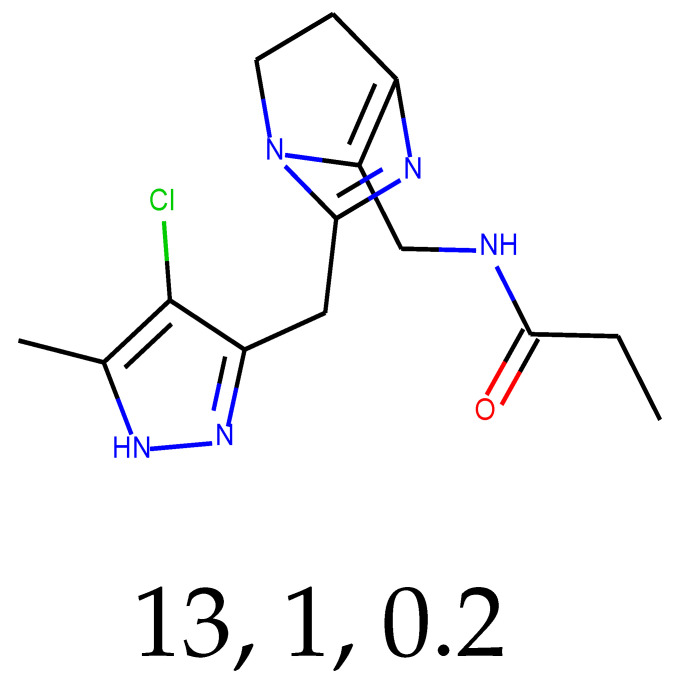	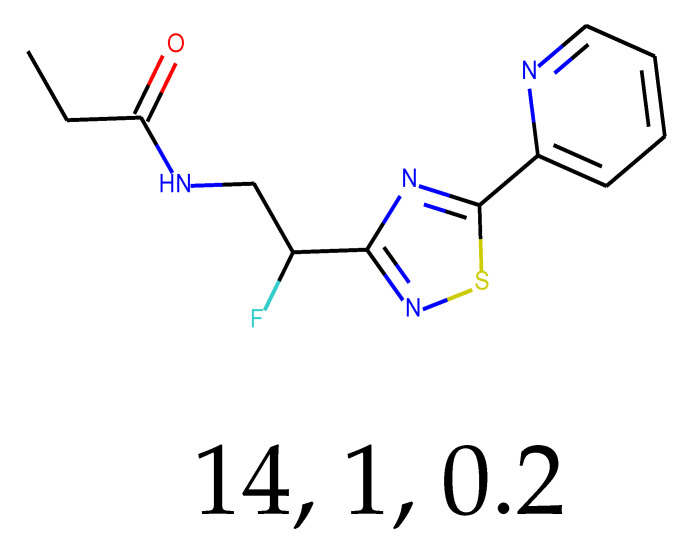	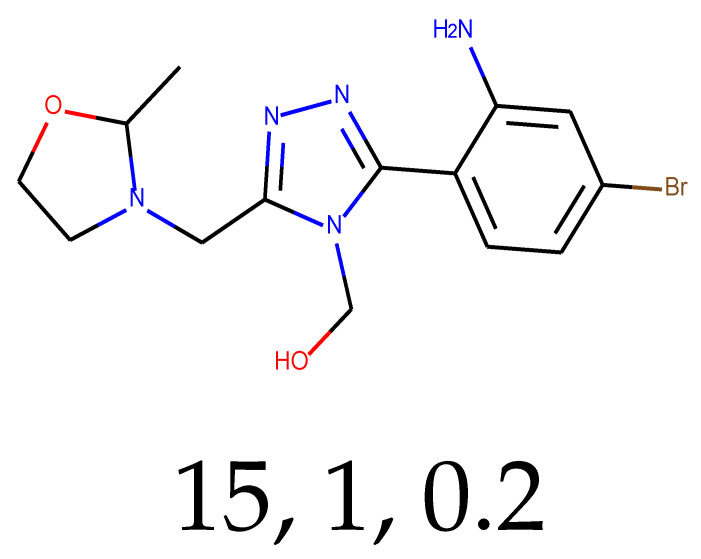	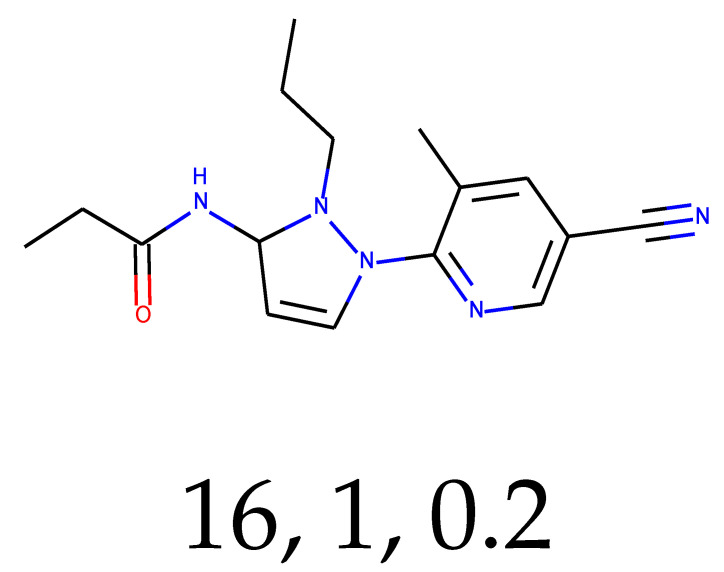
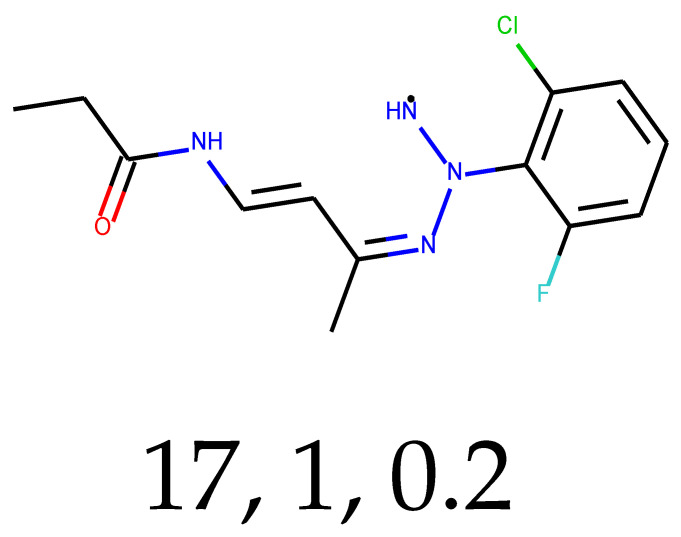	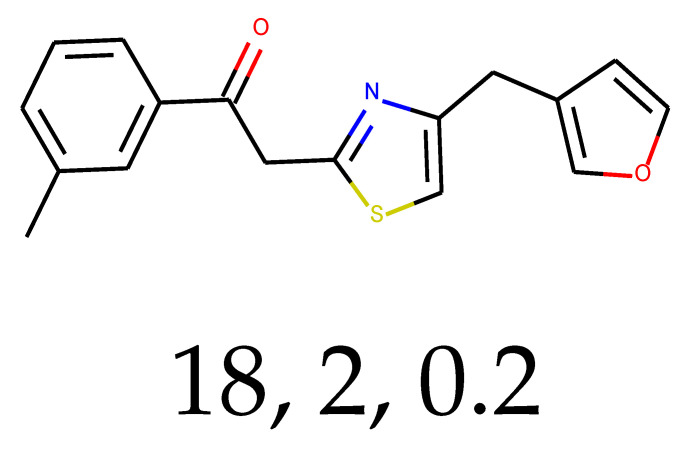	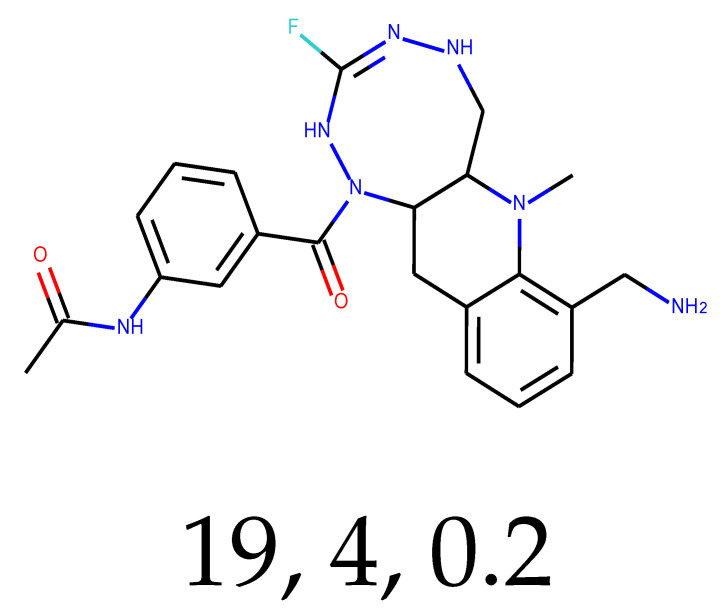	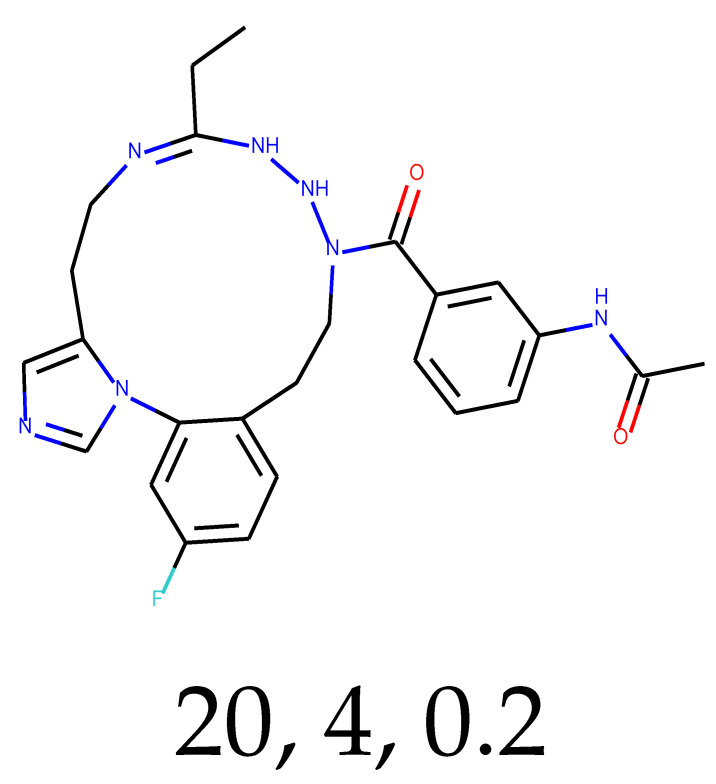

**Table 2 ijms-24-01762-t002:** “My score”, QED normalized, and SYBA scores normalized, see SM Equation (S3) for selected structures with Appendix A; all structures meet the QED threshold and Lipinski’s rule of 5. They are used to determine the normalized QED and SYBA scores and “My score” (see Appendix A).

Molecule Number	QED	QED Normalized	SYBA Score	SYBA Score Normalized	“My Score”
4	0.82	0.76	39.47	0.75	75.10
5	0.62	0.22	5.24	0.64	42.70
6	0.88	0.92	35.55	0.73	82.67
7	0.54	0.02	79.24	0.87	44.65
8	0.79	0.68	119.44	1.00	84.00
9	0.67	0.35	90.13	0.91	63.02
10	0.53	0.00	47.43	0.77	38.68
11	0.53	0.00	85.03	0.89	44.54
12	0.74	0.56	7.05	0.64	60.04
14	0.91	1.00	47.22	0.77	88.54
15	0.79	0.69	17.76	0.68	68.36
16	0.90	0.97	4.28	0.63	80.42
17	0.67	0.36	15.68	0.67	51.67
18	0.67	0.35	95.91	0.93	63.93
19	0.54	0.01	6.40	0.64	32.45
20	0.54	0.01	47.67	0.77	39.20

**Table 3 ijms-24-01762-t003:** Docked structures with docking scores in kcal/mol unit (see Appendix A).

Molecule Number	7NPC	7NP5	7KXD
4	−7.9	−8.2	−8.6
5	−6.5	−6.9	−6.7
6	−8.6	−8.1	−8.7
7	−9.2	−10.0	−8.7
8	−8.8	−9.0	−9.2
9	−8.0	−8.2	−8.5
10	−10.0	−9.1	−9.2
11	−8.7	−8.8	−10.0
12	−9.9	−9.5	−10.0
14	−6.6	−6.9	−7.2
15	−7.0	−7.1	−7.3
16	−7.5	−7.4	−7.7
17	−7.0	−7.1	−7.3
18	−9.7	−8.1	−8.3
19	−8.8	−10.0	−10.0
20	−9.5	−10.0	−9.6

**Table 4 ijms-24-01762-t004:** Results of molecular docking for structures found in the Protein Database’s raw files [33,34], along with QED and SYBA scores—see Appendix A.

Structure	QED	SYBA Score	Docking Result/Name of Domain
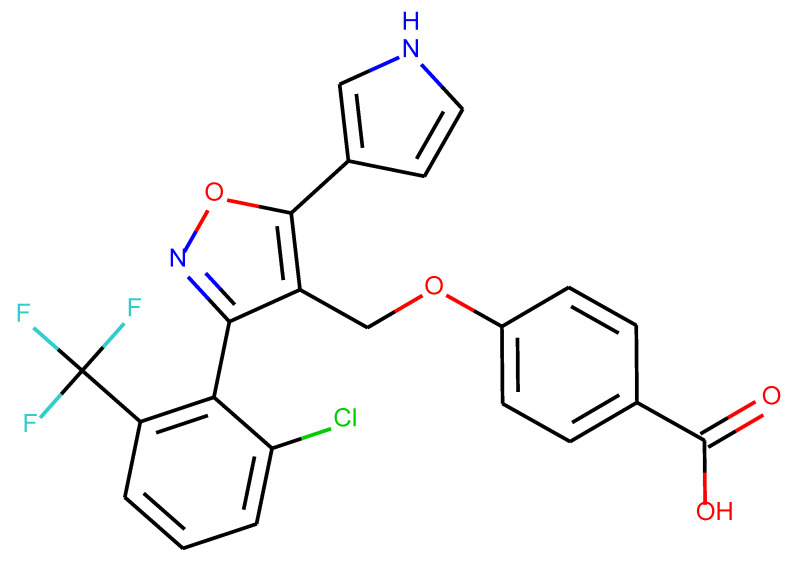	0.35	45.11	−13.84 kcal/mol/7NPC −14.19 kcal/mol/7NP5
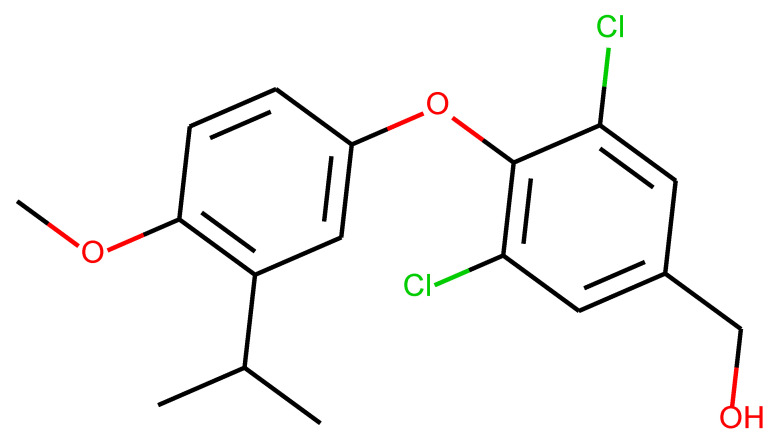	0.79	71.02	−8.66 kcal/mol/7KXD

**Table 5 ijms-24-01762-t005:** These structures were used as prediction initializers.

Structure	QED	QED Normalized	SYBA Score	SYBA Score Normalized	“My Score”	RORγ Activity
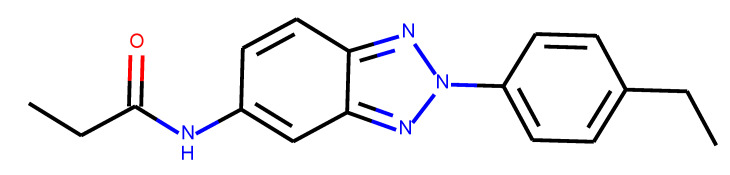	0.80	1.00	140.57	0.70	85.19	agonist ^1^
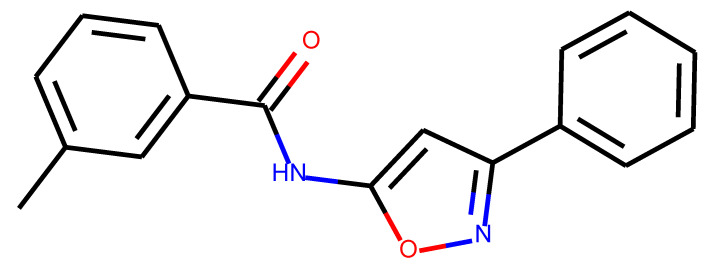	0.79	0.96	119.44	0.59	77.43	agonist ^1^
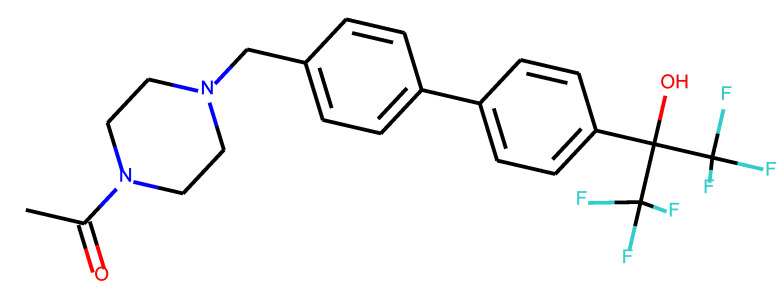	0.69	0.60	119.78	0.59	59.77	inverse agonist ^1^
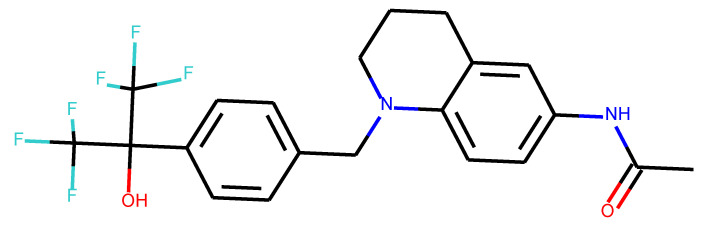	0.66	0.48	116.61	0.58	52.72	inverse agonist ^1^
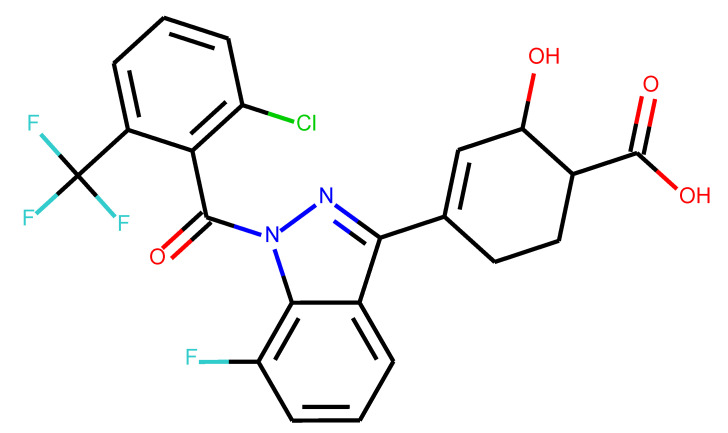	0.52	0.00	5.72	0.00	0.00	inverse agonist ^1^

^1^ RORγ activity is known from Zhang, et al. publication [20].

## Data Availability

All publication-related information can be accessed at this address: https://github.com/XDamianX-coder/seq_to_seq_and_dock_AMU (accessed on 10 December 2022). Appendix A can also be downloaded using the link provided above.

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
