# Peer review of "Neural Networks in the Design of Molecules with Affinity to Selected Protein Domains"

_ijms, 2023, doi:10.3390/ijms24021762_

Round 1

Reviewer 1 Report

The study entitled Neural networks in the design of molecules with affinity to selected protein domains proposed a method for identifying new drug molecules against selected protein domains. The method developed by combining artificial intelligence, especially the neural network approach and generated new potential drugs, and performed molecular docking analysis to check the binding efficiency of the new ligands with the receptors. We believe the work is technically sound and scientifically valid. In order to be understood by the general readership, we would like to suggest a few additions to the manuscript.

1.      How can we assume the compounds predicted using AI will also be biologically active as they are derived from the known molecules? Authors are advised to provide more insights into this aspect.

2.      Although the paper mainly discusses the method, it would be better to discuss the biological significance of the proteins selected for the study.

3.      The authors have focused on the preparation of 121,000 structures for a model. On what basis is the structure differentiated and used as training and validation data?

4.      The compounds are extracted in SMILES format, converted to neural network favoured SELFIES format, and then converted back to SMILES. Authors are advised to describe each format and the significance of representing the chemical structure in a specific format.

5.      What is the reason for selecting RORγ family protein receptors? Can the method be reproduced for any other protein families with known domains?

6.      The authors mentioned that the method's main limitation is the molecular sequence's length. In this scenario, we may lose the potential drug leads that effectively work against the receptors.

7.      Using natural compounds to develop new drugs is gaining the scientific community's attention. Natural compounds are generally larger molecules. Can the method predict the biologically active compound from the natural compounds?

8.      Molecular dynamic simulations (MDS) contribute effective role in molecular biology and new drug discovery process. If there is room for improvement of the current method, the authors are advised to look for the MDS analysis of the newly predicted compounds complex with the receptors.

Author Response

Dear Reviewer,

Thank you for your constructive comments.

I have included responses to the comments below.

Ad 1. How can we assume the compounds predicted using AI will also be biologically active as they are derived from the known molecules? Authors are advised to provide more insights into this aspect.

It can be assumed based on the molecular docking technique and partial resemblance to existing drugs with proven biological action. The proposed here technique might be considered as a random search in the neighborhood of particular species, e.g. the one for which the biological activity is experimentally known. The variations introduced into the species, to large extent keep the pharmacophoric nature of the initial molecule. Based on this presumption one can initially assume that the derived species will pose similar properties. Still, experimental validation is the best way for an ultimate resolution.

The appropriate description was provided at the end of Section 2.5.

“The biological activity of the derived structures can be assumed based on the molecular docking technique and a partial resemblance to existing drugs with confirmed biological action. The technique suggested here could be thought of as a random search in the region of a specific species, such as one whose biological activity has been experimentally determined. The changes incorporated into the species preserve the pharmacophoric properties of the original molecule to a considerable extent. Based on this assumption, it is reasonable to expect that the derived species will have similar characteristics. Nonetheless, experimental validation is the best technique to get a conclusion.”

Ad 2. Although the paper mainly discusses the method, it would be better to discuss the biological significance of the proteins selected for the study.

The biological significance is that they are linked to many human diseases like atherosclerosis, osteoporosis, autoimmunological disorders, obesity, asthma, and cancer. RORγ are considered a master regulator of Th17 differentiation.

We have extended the Introduction to provide more biological insight into the importance of the nuclear receptor.

“The protein domains researched in this study belong to ROR$\gamma$ proteins. They are referred to as orphan receptors since their natural ligands are undetermined \cite{hummasti_adopting_2008}. ROR$\gamma$ proteins are associated with several processes in our bodies, including metabolic regulation, whole-body development, cell apoptosis, homeostasis maintenance, and circadian rhythm modulation. \cite{jetten_retinoid-related_2009, giguere_orphan_1999}. The biological relevance is that they are associated with a variety of human diseases such as atherosclerosis, osteoporosis, autoimmunological disorders, obesity, asthma, and cancer \cite{zhang_ror_2015, kurebayashi_retinoid-related_2000, billon_inhibition_2016}. RORs are thought to be the key regulators of Th17 differentiation \cite{zhang_increasing_2012, solt_suppression_2011}. They can be found in our bodies' heart, liver, testis, and muscles \cite{jetten_retinoid-related_2009, medvedev_cloning_1996}.”

Ad 3. The authors have focused on the preparation of 121,000 structures for a model. On what basis is the structure differentiated and used as training and validation data?

The training and validation data were sampled from the ZINC20 tranches. The structures were chosen based on the sklearn functionality “train test split,” which helps to divide a specific proportion of data into training and validation sets. There is no specific data preparation for differentiation for training and validation sets.

Please see Section 2.1:

“The data was then divided into training and validation sets using the sklearn library's "train test split" functionality. There was no preparation of specific data for differentiation of the training and validation sets. The 10% of the acquired data set has been used as validation data.”

Ad 4. The compounds are extracted in SMILES format, converted to neural network favoured SELFIES format, and then converted back to SMILES. Authors are advised to describe each format and the significance of representing the chemical structure in a specific format.

The SMILES format is quite useful for Python molecular docking techniques, and it is well supported by more widely used libraries such as RDKit and pubchempy. The SELFIES format has been examined for neural network applications to see if improper, semantically wrong, SELFIES production is possible. The SELFIES format is relatively new, and it has only been tested for in-silico purposes.

The appropriate information was provided at the end of Section 2.3.

“The SMILES format is very useful for Python molecular docking approaches, and it is effectively supported by popular libraries like RDKit or PubChemPy. The SELFIES format has been investigated for neural network applications and to determine whether semantically incorrect SELFIES creation is possible.”

Ad 5. What is the reason for selecting RORγ family protein receptors? Can the method be reproduced for any other protein families with known domains?

The biological significance is that they are linked to many human diseases like atherosclerosis, osteoporosis, autoimmunological disorders, obesity, asthma, and cancer. RORγ are considered a master regulator of Th17 differentiation. The process can be reproduced using any known protein domains and any known chemicals with activity against them.

The appropriate information was provided at the end of the Conclusions (4).

“Any known protein domain and any known chemical having activity against it can be used to replicate the process.”

Ad 6. The authors mentioned that the method's main limitation is the molecular sequence's length. In this scenario, we may lose the potential drug leads that effectively work against the receptors.

The neural network used here can handle a maximum of 65 SELFIES characters, however, there is no limit to the maximum number of SELFIES representations of a molecule in general. It is determined by the neural network's input layer definition. This technique can handle SELFIES representations of molecules with arbitrarily chosen lengths, but the treatment of long representations will be computationally expensive. To summarize, this particular constraint applies to this specific model architecture, in general, the real limiting factor is the computer power required for longer SELFIES vectorization and neural processing.

The appropriate information was provided at the end of Section 3.1

“The main limitation of the used model is the maximal length of molecular sequence that can be effectively encoded. The neural network employed can handle up to 65 SELFIES characters, although there is no restriction to the maximum SELFIES length in general. The neural network's input layer definition determines it. This approach can handle SELFIES representations of molecules of any length, however, dealing with lengthy representations will be computationally expensive.”

Ad 7. Using natural compounds to develop new drugs is gaining the scientific community's attention. Natural compounds are generally larger molecules. Can the method predict the biologically active compound from the natural compounds?

This approach can handle SELFIES representations of molecules of any length, however, dealing with lengthy representations will be computationally expensive. It all depends on what data are used to train the neural network. Any structure's vectorized representation can be used as an input. Please keep in mind that biological activity requires experimental verification. This method can only demonstrate the potential affinity for specific protein domains.

Ad 8. Molecular dynamic simulations (MDS) contribute effective role in molecular biology and new drug discovery process. If there is room for improvement of the current method, the authors are advised to look for the MDS analysis of the newly predicted compounds complex with the receptors.

MDSs were not included in the study. It would undoubtedly improve the method provided. However, molecular dynamic simulations were not investigated in the scope of current research. However, we are thankful for these suggestions, we will consider that as an avenue for future scientific efforts.

We have provided an appropriate statement at the end of Section 3.5.

“Molecular dynamic simulations (MDS) have a significant impact on molecular biology and the discovery of novel drugs. The study did not include this. It is a location for undeniable improvements to the method presented.”

Kindest regards,

Damian Nowak

Reviewer 2 Report

The manuscript entitled " Neural networks in the design of molecules with affinity to selected protein domains" reports a model that can create molecules based on Neural networks (AI) for further molecular modelling. The study presents an interesting application of artificial intelligence on biological chemistry. Thus, I suggest that the manuscript is suitable for publication after minor revision.

Comments:

-          Authors need to emphasize on the proeblem that the model could sovle.

-          Abbreviation should be defined then abbreviated through the manuscript, for example RORγ.

-          Authors should  defined “compounds with the SELFIES code length in the range of 30 to 50” in chemical structure terms ( M.W. range, numbe of atoms, ..etc).

-          The importantce of each of three classifier should be mention in the main text ( in addtion to the difference between them or the added value of each claasifer over the preceding one).

-          Authors are asked to clarifiy fif the model can role out the decoys that have no activity on the selected target and how?

Author Response

Dear Reviewer,

Thank you for your constructive comments.

I have included responses to the comments below.

Ad 1. Authors need to emphasize on the proeblem that the model could sovle

Based on the initial structures, the model may predict new chemical structures. Depending on the training data, the model can be employed in drug design or chemical design in general.

We have elaborated the Introduction to fulfill this expectation. Please see Section 1.

“The model may predict new chemical structures based on the initial structures. The model can be used in drug design or chemical design in general, depending on the training data. This strategy can be used to solve a variety of structural design problems.”

Ad 2. Abbreviation should be defined then abbreviated through the manuscript, for example RORγ

We have defined all used abbreviations.

Ad 3. Authors should  defined “compounds with the SELFIES code length in the range of 30 to 50” in chemical structure terms ( M.W. range, numbe of atoms, ..etc).

 In chemical structure terminology, it refers to the length of chemical structures whose length in the SELFIES representation ranges from 30 to 50. It is difficult to attribute it definitively to molecular characteristics. However, it can be viewed in certain ways as the number of atoms in a specific molecule.

Please see the middle of Section 2.1

“The length of SELFIES is proportional to the number of atoms in a specific structure.”

Ad 4. The importantce of each of three classifier should be mention in the main text ( in addtion to the difference between them or the added value of each claasifer over the preceding one).

The significance of each classifier is explained here and in the manuscript from now on.

We can use the QED descriptor classifier to assess the potential Drug-Likeness of projected structures.

Lipinski's rule of 5 also allows us to test predicted structures for Drug-Likeness.

The SYBA classifier checks the structure's synthesis accessibility.

All of these are utilized as classifiers before molecular docking, allowing for the selection of the "best" structures for in vitro molecular docking.

Please see the ending of Section 2.5.

“To analyze the potential Drug-Likeness of proposed structures, the QED and Lipinski's rule of 5 classifiers are utilized. The third classifier checks the structure's accessibility to synthesis. They are all employed before the molecular docking technique, assisting in the selection of the "best" structures for further testing.”

Ad 5. Authors are asked to clarifiy fif the model can role out the decoys that have no activity on the selected target and how?

As proposed here neural network is not capable of determining whether a certain structure is active or not. This could be accomplished by an application of subsequent classifiers oriented on the biological activity of certain receptors, e.g. nuclear receptors. The molecular docking score can also be used as a classifier of potential affinity to selected nuclear receptors. The molecular dynamic simulations and subsequent binding free energy calculation could also shed some light on the potential susceptibility of particular species to considered receptors. However, this was not in the scope of the presented research, but might be considered as a future direction of the scientific efforts.

Please see the very end of Section 3.1

“The neural network proposed here is incapable of distinguishing whether a specific structure is active or not. This could be performed by employing subsequent classifiers and molecular docking based on the biological activity of specific receptors, such as nuclear receptors.”

Kindest regards,

Damian Nowak

Round 2

Reviewer 1 Report

Revisions are satisfactory, the authors are advised to check the language while they check their proof.